# Bioengineering Strategies to Create 3D Cardiac Constructs from Human Induced Pluripotent Stem Cells

**DOI:** 10.3390/bioengineering9040168

**Published:** 2022-04-10

**Authors:** Fahimeh Varzideh, Pasquale Mone, Gaetano Santulli

**Affiliations:** 1Department of Medicine, Wilf Family Cardiovascular Research Institute, Einstein-Mount Sinai Diabetes Research Center (*ES-DRC*), Einstein Institute for Aging Research, Albert Einstein College of Medicine, New York, NY 10461, USA; fahimeh.varzideh@einsteinmed.edu (F.V.); pasquale.mone@einsteinmed.edu (P.M.); 2Department of Molecular Pharmacology, Fleischer Institute for Diabetes and Metabolism (*FIDAM*), Einstein Institute for Neuroimmunology and Inflammation (*INI*), Albert Einstein College of Medicine, New York, NY 10461, USA

**Keywords:** 3D bioprinting, 3D models, cardiac organoid, cardiac tissue engineering, cardiomyocytes, CPVT, differentiation, disease modeling, drug development, drug screening, ECM, heart-on-a-chip, hiPSC-CMs, human induced pluripotent stem cell, iPSC, Layer-by-Layer, maturation

## Abstract

Human induced pluripotent stem cells (hiPSCs) can be used to generate various cell types in the human body. Hence, hiPSC-derived cardiomyocytes (hiPSC-CMs) represent a significant cell source for disease modeling, drug testing, and regenerative medicine. The immaturity of hiPSC-CMs in two-dimensional (2D) culture limit their applications. Cardiac tissue engineering provides a new promise for both basic and clinical research. Advanced bioengineered cardiac *in vitro* models can create contractile structures that serve as exquisite *in vitro* heart microtissues for drug testing and disease modeling, thereby promoting the identification of better treatments for cardiovascular disorders. In this review, we will introduce recent advances of bioengineering technologies to produce *in vitro* cardiac tissues derived from hiPSCs.

## 1. Introduction

Cardiovascular diseases (CVD) remain the leading cause of mortality worldwide [1]. According to statistics of the American Heart Association, there are more than 2300 deaths from CVD each day; on average, someone dies of CVD every 36 seconds in the United States. Every year, the United States spend an estimated $363.4 billion on CVD and it is projected that these costs will increase to 1044 billion dollars by 2030 [2].

The differentiation of induced pluripotent stem cells (iPSCs) into cardiac cells over the past decade has provided a powerful tool for cardiovascular research [2]. In 2006, Takahashi and Yamanaka reported, in a seminal paper, the possibility to induce iPSCs through reprogramming adult fibroblast by c-Myc, Oct3/4, Sox2, and Klf4 transcription factors [3]. Since then, iPSCs have been shown to differentiate into any cell type of human body including neurons, hepatocytes, and cardiomyocytes (CMs). Thus, iPSC technology provides a great opportunity to produce patient specific cell lines that can be used to study mechanisms of diseases, drug discovery, and therapeutic testing *in vitro* [4,5]. Patient- specific iPSCs-derived CMs (hiPSC-CMs) can mimic aspects of genetic diseases, metabolic dysfunction, and drug responses. However, hiPSC-CMs show properties similar to fetal human CM rather than adult CM [6]. The process of testing new drugs for safety and efficacy is time consuming and expensive [7,8]. These procedures are often carried out on two-dimensional (2D) cell culture and animal models, which are unable to properly reproduce the human physiological conditions [9]. In 2D *in vitro* models, single cell types are used, which do not accurately mimic functional multicellular tissue *in vivo*. A three-dimensional (3D) culture environment is necessary to obtain a proper CM phenotype, in a way as similar as possible to *in vivo* cardiac cells.

Henceforward, there is a need to improve CM differentiation in order to properly recapitulate the physiological and pathological characteristics of adult heart cells. 3D culture and engineering strategies can overcome 2D culture limitations. A combination of iPSC with 3D culture models can better replicate an *in vivo* microenvironment, including cell–cell and cell–matrix interactions, chemical, mechanical properties, and drug activities. The physiological immaturity of CMs is attenuated using combination iPSC technology with advancements in engineered biomaterials, such as microelectromechanical systems devices, heart-on-chip, bioprinting, and organoid technology [10,11,12]. In this review, we will summarize the most updated bioengineering approaches harnessed to generate *in vitro* cardiac tissues (Figure 1).

## 2. 3D Culture Using Scaffold-Free and Scaffold-Based Approaches

The generation of mature human CMs is a focal challenge in cardiovascular research. Unfortunately, CMs differentiated from hiPSCs in 2D conditions are immature in terms of morphology, expression of sarcomeric proteins, metabolism, and electrophysiology properties, such as excitation–contraction coupling (ECC) and contraction velocity [13]. Adult CMs are elongated, multinucleated, rod-shaped with organized sarcomeres, and exhibit a well-developed calcium (Ca^2+^) handling system, T-tubule organization, and action potential (AP) [14]. 

In contrast, hiPSC-CMs are mononucleated with disorganized sarcomeres and immature in terms of electrical coupling, AP, and gene expression (*MYH6 > MYH7*, *TNNI > TNNI3*, low expression of electrophysiological channels including *KCNJ2*, *KCNJ8, KCNH7*, *SCN5A*, *HCN4*, *GJA1*, *CACNA1C*, *RYR2*, *ATP2A2*, *CASQ2*, and *CAV3*) [15,16,17,18]. Small molecule-based approaches allow the obtainment of healthy hiPSC-CMs and patient specific hiPSC-CMs in small-scale and large-scale. Temporal modulation of Wnt/β-catenin signaling by the GSK3 inhibitor CHIR99021 (leading to an activation of Wnt signaling) or IWR1 or IWP2 (inhibitors of Wnt signaling) orchestrate CM differentiation [19,20,21,22]. Other studies demonstrated that 3D culture of hiPSC-CMs increased maturation as indicated by organized sarcomere structures, a rise in *MYH7/MYH6* and *TNNI3/TNNI1* ratio, potassium (K^+^) channels such as *KCHJ2*, and a metabolic switch from glucose or lactate to fatty acids [23,24,25,26,27].

There are two main methods for 3D culture of hiPSC-CM: scaffold-free and scaffold-based. In the scaffold-free method, hiPSC-CMs are cultured by aggregation through spinner flasks, hanging drop, non-adhesive U-shaped wells, V-bottom 96 well microplates, agarose microwells, and agitation culture; spherical microtissues are formed by hiPSC-CM alone or in co-culture with stromal cells [28,29]. In the scaffold-based method, hiPSC-CMs are encapsulated into extracellular matrix (ECM), natural hydrogels (fibrin, collagen, gelatin, and chitosan), or synthetic hydrogels (polycaprolactone, polyethylene glycol, and poly vinyl alcohol) [28,30,31,32,33,34,35,36].

ECM composition has been determined as one of the instrumental factors necessary for the maturation of neonatal CM by affecting the assembly of myofibrils [37]. Decellularized ECM obtained from the heart offers a network of glycosaminoglycans, proteoglycans, and other proteins that are present in native tissues, representing ideal biochemical and mechanical characteristics for proliferation and differentiation of cardiac cells, expression of cardiac markers of cardiac progenitor cells, and ultimately cardiac regeneration [38]. 

Hydrogels derived from decellularized tissues mimicking the properties of the native extracellular matrix can create a natural tissue microenvironment to support cellular processes [39,40,41]. Various studies showed that culturing hiPSC-CMs on ECM increased force contraction, cellular organization, and expression of late markers [42,43,44,45]. Decellularized ECMs are used to generate cardiac patches [46]. Decellularized tissues have already been tested in therapeutic applications such as heart valve implantation [47]. There is a high viability of hiPSC-CM within ECM or other hydrogels, which is very useful for cardiac drug testing [48,49].

The combination of 3D culture with other approaches, including co-culture with non-myocytes (endothelial cells, fibroblasts, and mesenchymal cells) [15,30,50,51,52,53,54,55,56], mechanical [57,58,59] and electrical [60,61] stimulation, addition of hormones like insulin-like growth factor-1 (IGF-1), or thyroid hormone, [14,62,63], could enhance hiPSC-CMs maturity (Figure 2).

The molecular interactions between endothelial cells, fibroblasts, and vascular smooth muscle cells (VSMCs) are essential for CM maturation and contractility as well as disease modelling. Indeed, co-culturing hiPSC-CMs with fibroblasts improved contractility [28] and several studies demonstrated that the overexpression of *KCNJ2* [64], *CASQ* [65], and a combination of miRNAs (e.g., miR-125b, miR-199a, miR-221, and miR-222) [66] increased maturation properties of CMs; for instance, in terms of hyperpolarized resting membrane potential and Ca^2+^ handling. Treatment of hiPSC-CMs with triiodothyronine (T3) increased cell size, organization of sarcomere, contraction, mitochondrial biogenesis, and reduced proliferation [62]. Additionally, a combination of thyroid hormone, dexamethasone, and insulin-like growth factor-1 (TDI) enhanced sarcomeric alignment, Ca^2+^-transient kinetics, expression of ion channels (*KCNJ2*, *SCN1B* and *RYR2*), density of mitochondria, and T-tubule formation [67]. Recently, Anna Skorska and coworkers used super-resolution microscopy and deep machine learning to evaluate the quality of sarcomere network in hiPSC-CM; to improve the maturity of CMs, prolonged *in vitro* culture, treatment with thyroid hormone and dexamethasone, and micro patterned surfaces were applied, detecting an increase in sarcomere density, sarcomere length, and sarcomere alignment [68]. Equally important, in order to model long QT syndrome type 3 (LQT3) and dilated cardiomyopathy with mutation in *SCNA5A* and *RBM20*, respectively, Feyen and collaborators developed a maturation medium consisting of DMEM (free glucose), albuMAX, L-carnitine, taurine, creatine, and ascorbic acid; this medium increased CM maturation in 2D and 3D cultures, as noted by reduction of beating rate and proliferation, high fatty acid uptake and respiration rates, enhanced inward rectifier K^+^ current (I_K1_), diastolic membrane potential, upstroke velocity of action potential (AP), and Ca^2+^ handling [69].

Bioengineering approaches are used to generate engineered heart tissue (EHT) (ring-shaped, cylindrical, or longitudinal structures) using ECM, hydrogels, scaffolds, and microwell molds [70]. Culturing EHT in a bioreactor with electrical stimulation enhances the expression of cardiac specific proteins [71]. Electrical stimulation of a 3D culture of hiPSC-CMs with human fibroblasts in fibrin hydrogel upregulated gene expression of *RYR2, ITPR3, KCNH2, MYH7,* and *CAV3* [59]. Electrical stimulation can increase the protein expression of Cx43, N-cadherin (N-Cad), and ZO-1, which improve the interaction between cells and release of Ca^2+^ [72,73]. Joseph Wu’s team generated an engineered heart muscle (EHM) by mixing human iPSC-CMs with human IMR-90 fibroblasts in collagen hydrogel under different stretching conditions [57]; EHMs exhibited upregulation of *CAV* (*caveolin-3*), *KCNJ2*, TNNT2, β-adrenergic receptors (*ADRB1* and *ADRB2*), and the Ca^2+^ Voltage-Gated Channel Subunit Alpha1 C (*CACNA1C*), with a reduction of beating rate and a higher degree of Ca^2+^ amplitude [57]. Willem De Lange and colleagues introduced a novel model in which hiPSC-CMs and hiPSC-derived cardiac fibroblasts were cocultured in a 3D fibrin matrix to form engineered cardiac tissue constructs (hiPSC-ECTs), which are responsive to well-established physiological stimuli, including β-adrenergic stimulation, and stretch, and display Ca^2+^-handling and contractile kinetics that are similar to the human myocardium [74]. The combination of a PPARα agonist, dexamethasone, T3 and palmitate in media containing low glucose was shown to be very effective at inducing a mature phenotype in human pluripotent stem cell (hPSC)-derived ventricular CMs, increasing the expression of CX43 [75]. Recent work from the lab led by Jan Buikema elegantly illustrates that there is a complex biology behind the cellular and nuclear division of mono- and bi-nuclear CMs, with a shared-phenomenon of sarcomere disassembly during mitosis [76].

Innovative engineering approaches have made remarkable contributions to promote CM maturation. In the following section, we discuss some advanced technologies to create cardiac tissue.

## 3. Cardiac Organoids

The intrinsic capability of pluripotent stem cell (PSC) to discriminate various cell forms that are able to self-organize has implemented the possibility to generate ‘organs-in-a-dish’ known as organoids [77]. The word “organoid”, meaning ‘resembling an organ’, was first introduced by Smith and Cochrane [78] in 1946 to define a cystic teratoma. The term organoid refers to a 3D self-organized structure containing several cell types (organ specific) resulting from stem cells that summarize the *in vivo* organ’s architecture and function [79,80,81,82]. Since the organoid creation is a tremendous breakthrough in biological research, scientists are particularly interested in developing advancements in this model system to produce 3D tissues that can better mimic the organ’s physiology. The PSC and ASC (adult stem cells) generate the organoid and each of these components has specific applications. Organoids represent ideal suitable tools for disease modeling, regenerative medicine, and drug testing [83,84,85,86].

The first PSC-derived organoids were produced for the brain [87] and then other organoids have been generated for many other organs including optic cup [88,89], lung [90], kidney [91], liver [92,93], intestine [94,95], and stomach [96,97]. Despite extensive research on a variety of organoids, cardiac organoids have been less examined [98]. Cardiac organoids can reflect *in vivo* cardiogenesis and cardiac organoid models can ideally help us to better understand the mechanism of heart disease [70,99]. Organoids derived from patient-specific or genome-edited human iPSC lines provide a personalized platform for drug testing, disease modeling, and organ transplant [100].

There are two main methods to generate cardiac organoids, directed assembly and self-organization (also known as self-assembly). In the directed assembly method, CMs, endothelial cells, VSMCs, and fibroblasts are aggregated utilizing hydrogels, Matrigel or ECM [9,98,101,102]. In the second approach, hPSCs are triggered toward cardiac lineage after spheroid formation [103,104,105,106,107]. Direct assembly of cardiac organoids has been reported from co-culturing of human embryonic stem cells (hESC)-derived cardiac progenitor cells (hESC-CPC), endothelial cells, and mesenchymal stem cells, leading to improved maturation of CMs in terms of the expression of proteins involved in the typical CM structure and some key ion channel genes [108,109]. After co-aggregation, endothelial cells form vessel-like structures within the cardiac organoid.

One of the main problems of organoids is hypoxia in their central portion [110,111,112]. This issue can be disentangled by favoring the formation of vessel structures within spheroids and organoids. Vascularization of 3D hiPSC-CM tissue is essential for *in vitro* and *in vivo* survival [113,114,115] and can be achieved by providing an exquisite combination of growth factors, cytokines, and adhesion molecules triggering proliferation, differentiation, and regeneration of endothelial cells into 3D structures [116,117,118,119,120,121,122].

The LEFTY-PITX2 signaling pathway plays crucial roles in the maturation of cardiac organoids derived from hiPSC-cardiac mesoderm [123,124]. Using a specific cocktail of growth factors, Sasha Mendjan and coworkers were able to generate self-organizing cardioids using seeding hESCs and hiPSCs in 96-well plates (3D culture) followed by cardiac differentiation; the authors demonstrated that cavity morphogenesis is controlled by the mesodermal Wnt-BMP signaling axis [107], proving that cardioids represent a suitable platform to study mechanisms of cardiogenesis and heart diseases. 2-Cl-C.OXT-A (COA-Cl) increased contraction of cardiac organoid generated by co-culturing human dermal fibroblasts (HDFBs) with hiPSC- endothelial cells and hiPSC-CMs onto a 96-well plate with a spindle-shaped bottom [125]. As mentioned above, there is an ongoing challenge to recapitulate postnatal maturation of hPSC-CMs, overcoming an important limitation of their application for cell therapy or drug discovery. Mills and collaborators generated cardiac organoids by culturing hPSC-CMs and fibroblasts within heart-dyno platform; they found that key proliferation pathways including β-catenin and Yes-associated protein 1 (YAP1) were repressed during maturation [126]. The same group screened 5000 compounds on cardiac organoids to find activators of CM proliferation, identifying the mevalonate pathway as fundamental to reenter the cell cycle, both *in vivo* and *in vitro* [127]. In a recent study, Todd McDevitt’s research group described multilineage organoids emphasizing how the presence of gut tissue in an organoid provides paracrine factors and enhances CM maturation within hiPSC- organoids [128]. Similarly, Elisa Giacomelli and associates verified the relevance of cardiac fibroblasts in the maturation of 3D cardiac microtissues [56]: maturation of hiPSC-CMs was markedly promoted by culturing them with hiPSC endothelial cells and hiPSC-cardiac fibroblasts, with high reproducibility across lines, batches, and samples [56]. Taken together, all these studies demonstrate that organoid technology is a powerful tool both in biological studies and for clinical applications.

## 4. 3D Bioprinting

3D printing and bioprinting technologies have emerged as promising methods for the production of functional tissue and organ regeneration [129]. Bioprinting creates cardiac models by recapitulating *in vivo* cell structure, geometry, and chemical and physiomechanical properties through a precise spatial control of cells and biomaterials [130]. To mimic the native structure of heart tissue, the most used biomaterials in bioprinting are synthetic or natural hydrogel or decellularized matrices such as collagen, fibronectin, and gelatin [131]. Researchers have attempted to vascularize *in vitro* cardiac tissue structures by 3D printers and a Layer-by-Layer approaches [132,133]. Layer-by-Layer techniques are used to preserve cell-cell and cell-ECM interactions [134,135,136,137]. For instance, Yuto Amano and coworkers developed vascularized hiPSC-CM tissue by applying a filtration-Layer-by-Layer technique [138].

To support contraction and vascularization, normal human cardiac fibroblasts and endothelial cells have been introduced into the 3D hiPSC-CMs tissue together with fibronectin and collagen coating [138,139]. Schaefer and collaborators have generated a bi-layer patch composed of hiPSC-CMs, endothelial cells, and pericytes; the bi-layer approach revealed an increase in force production, maturation, and viability compared to hiPSC-CMs alone [140]. Using bioprinting, Narutoshi Hibino and colleagues were able to generate biomaterial-free 3D cardiac patches: cardiac spheroids were generated by co-culture hiPSC-CMs with endothelial cells and fibroblast and 3D cardiac patches were fabricated by 3D printer; both CX-43 (as a gap junction protein) and CD31 (as a blood vessel marker) were observed in these structures [141]. Similarly, Khademhosseini’s research team obtained an endothelialized myocardium based on 3D printing and organ-on-a-chip; this group bioprinted microfibrous scaffolds consisting of bioink and endothelial cells and then CMs were seeded into an endothelialized scaffold; then, the endothelialized myocardium was embedded into microfluidic systems to screen CVD-related drugs [142].

## 5. Heart-on-a-Chip

3D cell culture presents some limitations concerning the use of hydrogels, natural extracellular matrices, synthetic polymers, microtissues, or organoids, including variation in tissue size and shape, as well as insufficient nutrient supply [143,144,145]; the lack of proper vascular perfusion and tissue–tissue interfaces, such as the interfaces between vascular endothelium, connective tissues, and stromal cells, represent other critical aspects [146]. Microfluidic technology has provided an opportunity to develop organ-on-a-chip (OOAC) and overcome these limitations. Microfluidic systems allow the supply of culture media and removal of debris cells [147]. The combination of 3D models with microfluidics creates complex multi-organ intercommunication via metabolite and chemical exchange [148]. These microfluidic minimal functional units can reflect structural and functional characteristics of human tissues, including key parameters such as shear stress and culture medium flow rate [149,150], pH level [151], and organ-tissue interactions [149,150,151]. Polydimethylsiloxane (PDMS) is often used to create microfluidic channels [152,153,154].

Milica Radisic and colleagues reported a new approach to engineering organ-on-a-chip; they used a synthetic polymeric elastomer, a scaffold named Angio Tube, to fabricate microchannels [155]. This system, known as AngioChip, allows the incorporation of organoids into organ-on-a-chip and supports perfusable vascular system [155]. Examples reported in recent years show that the heart-on-a-chip can be used to assess drug toxicity [36,131,142]. The combination of heart-on-chip with electrical or mechanical stimulation, or microenvironmental cues [60,156,157] resulted in enhanced maturation of cardiac tissues at the cellular and electrophysiological level. Of note, Sakai and associates [158] cultured hiPSC-CMs and rat sympathetic neurons in separate microchambers connected by microtunnels, confirming that CM beating can be controlled by sympathetic neurons [159], and that this method could be useful for evaluating sympathetic induced-cardiotoxicity in cardiac tissue [158]. In another study, myocardium-on-chip was created via a three-channel device fabrication: hiPSC-CMs were encapsulated into UV cross-linkable methacrylated gelatin (GelMA) and then seeded within the central channel of the chip; hiPSC-derived endothelial cells were cultured into two side channels to better replicate the *in vivo* microvasculature [160]. 

A cardiac microphysiological system was successfully generated through means of the fabrication of microfluidic devices and loading beating CMs derived from hiPSC within the central part of chip [161]. There were “endothelial-like” barriers between the cell chamber and nutrient channels to mimic the human vasculature [161]. Intriguingly, heteropolar biowire chips were made by seeding atrial and ventricular CMs to the opposite ends of microchannels; electrical stimulation of cells within the biowire effectively enhanced the maturation of CMs towards adult-like properties [162]. 

Other investigators generated humanized multi-tissue organ-on-chip platforms including liver, brain, lung, and cardiac tissues for systemic toxicity screening [163,164,165,166,167,168,169,170,171]. Despite significant progress in cardiac tissue engineering, we need to better assess its potential for disease modeling.

## 6. Cardiac Disease Modeling by 3D Engineering Tissue

3D models of hiPSC-CM provide a valuable platform for disease modeling, studying mechanism of diseases, drug testing, and toxicity screening, especially when compared to 2D cultures. Modeling of cardiac diseases including myocardial fibrosis, dilated cardiomyopathy, and LQT syndrome has been reported using stem cell biology and tissue engineering approaches [18,172]. The Seidman laboratory at Harvard generated cardiac tissue engineered by culturing hiPSC-CMs with mutations in titin-truncating variants and mesenchymal stem cells in micropatterned substrates; these structures recapitulated characteristics of patients with dilated cardiomyopathy properties, including decreased contractile force and disability of truncated protein to assemble with sarcomeric component [173]. Consistent with these observations, 3D cardiac structures with mutations in *PLN* [174] and *α-actinin 2 (ACTN2)* [175] genes have been reported. Most recently, Richards and associates developed a cardiac organoid model of myocardial infarction by adding norepinephrine; they demonstrated that these cardiac organoids exhibit properties generally observed following myocardial infarction, such as fibrosis and alterations in Ca^2+^ handling, and can also recapitulate *in vivo* cardiac responses to drugs [98,176].

## 7. Quality Control of hiPSC-CMs by Artificial Intelligence and Machine Learning

The functional assessment of hiPSC-CMs is mainly based on the evaluation of changes in morphology, structure, gene expression, and electrophysiological patterns. In the last years, researchers have proposed to assess the quality of hiPSC-CMs by advanced methods such as artificial intelligence and machine learning [177,178]. This screening is performed in the reprogramming stage of iPSCs to exclude abnormal cells and in the differentiation stage to select mature CMs for cell therapy applications [179].

Ca^2+^ transients play a central role in ECC and functionality of CMs [13,180]. Changes in Ca^2+^ cycling can be monitored by machine learning methods in order to find valid cell lines for drug testing and eventually improve disease diagnostics and treatment. For instance, Juhola and collaborators studied the abnormalities of Ca^2+^ transient signals in different CVD, including catecholaminergic polymorphic ventricular tachycardia (CPVT), long QT syndrome (LQT), and hypertrophic cardiomyopathy (HCM), by signal analysis and machine learning methods [181,182,183,184]. The effects of adrenaline and dantrolene on Ca^2+^ cycling properties of (CPVT)-specific iPSC-CMs have also been assessed by machine learning [185]. These machine learning techniques with iPSC-CMs provide an outstanding platform to diagnose cardiac disease and to better understand drug toxicity for CVD [186].

## 8. Conclusions and Future Directions

Significant progress in iPSC technology and hiPSC-CM generation has provided advances in understanding genetic and pathology of diseases. However, hiPSC-CM are immature in 2D culture. 3D cardiac models show improvement in CM maturation, as shown by their ability to reliably recapitulate *in vivo* cardiac microenvironment.

Bioprinting, heart-on-chip, organoids, and natural and synthetic scaffolds systems are used to generate 3D cardiac tissue. Although there is still a need to improve maturation and tissue vascularization, we are confident that enhancements in bioengineering approaches and the integration of rapidly advancing technologies will soon allow the generation of excellent systems for personalized medicine.

## Figures and Tables

**Figure 1 bioengineering-09-00168-f001:**
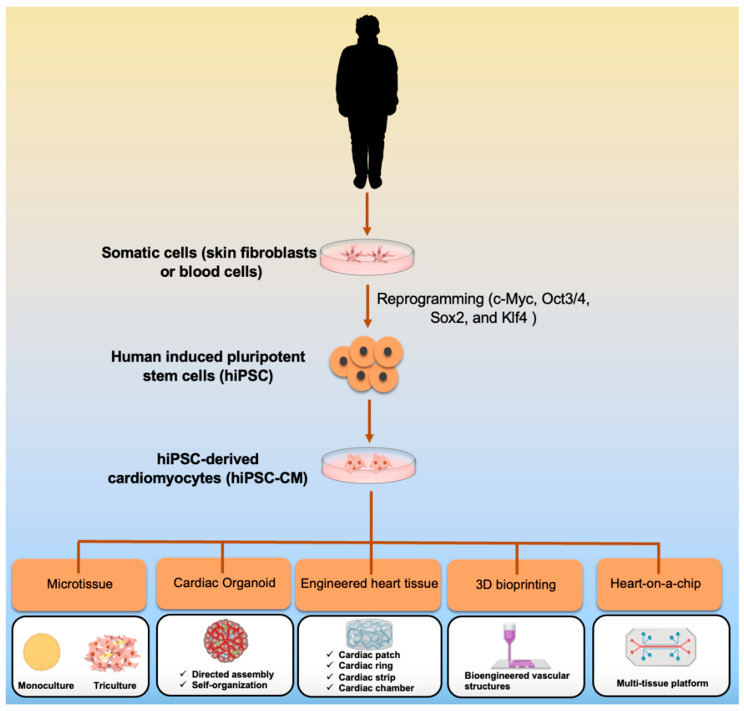
Tissue engineering approaches to create *in vitro* 3D cardiac tissue derived from hiPSCs.

**Figure 2 bioengineering-09-00168-f002:**
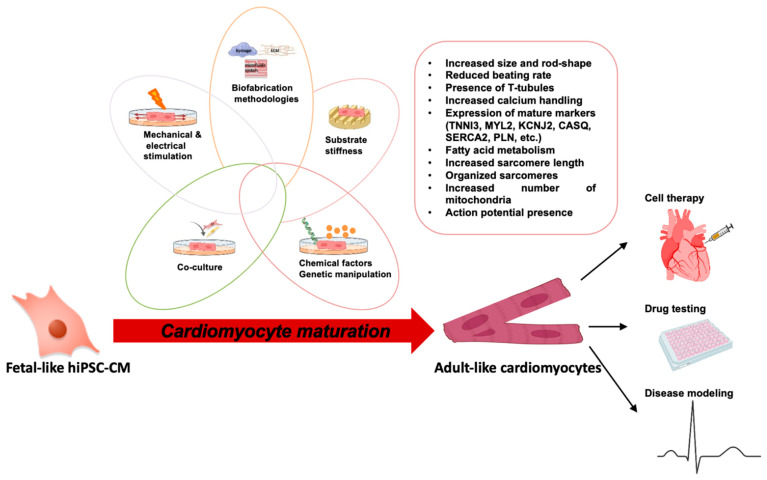
Strategies to increase maturation of hiPSC-CMs and their applications in basic research and in the clinical scenario. Electrical, mechanical, and biochemical factors, alongside genetic modifications and approaches aiming at adjusting substrate stiffness can be harnessed to enhance the maturation of hiPSC-CMs. Adult-like hiPSC-CMs are characterized by morphological, structural, genetic, and electrophysiological modifications.

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
