# Peer review of "Bioengineering Strategies to Create 3D Cardiac Constructs from Human Induced Pluripotent Stem Cells"

_bioengineering, 2022, doi:10.3390/bioengineering9040168_

Round 1

Reviewer 1 Report

Varzideh et al. review tissue engineering approaches to create cardiac tissues from induced pluripotent stem cells (iPSCs). Tridimensional cell models are presented as improved models to recapitulate cardiac phenotype and functions. The topic is very interesting; however, in its current form the review does not add anything new to existing reviews from other groups.

Major comments

Authors should extend all paragraphs and better cover the different facets of human cardiac tissue engineering, from the differentiation of iPSCs into CMs to CM patterning and functional maturation through the incorporation of selected biophysical and biochemical cues. How engineered cardiac constructs can help model and treat cardiac diseases should be also better discussed. Moreover, as the translational success of these constructs depends on the ability to recapitulate in vitro the complexity of functional heart, newest technological advances in molecular characterization of iPSC-CMs and prediction of their functions should be presented. There are several studies by Orita K., Juhola M., and Joutsijoki H., among others (recently reviewed by Coronnello C., Stem Cell Reviews and Reports, 2021), demonstrating the possibility to exploit artificial intelligence techniques to assess the quality of iPSC-CMs. Authors could spend a few sentences to discuss these techniques.

Furthermore, the review lacks of a critical evaluation. What is the authors’ view on the current status of cardiac tissue engineering? Do they have any recommendations for future research?

There are several recent papers that were not included in the review. Few examples are:

Dries A M Feyen, Cell Rep, 2020 Jul 21; Willem J de Lange, Am J Physiol Heart Circ Physiol, 2021 Apr 1; Qianliang Yuan, J Cardiovasc Dev Dis, 2022 Jan; Anna Skorska, Cell Mol Life Sci, 2022 Feb

Minor comments

Paragraphs are often disorganized with disconnected sentences. This is particularly evident for paragraph 2.

In paragraph 3, authors should spell out the acronym PSC.

Line 212. Correct "levl"

Author Response

Varzideh et al. review tissue engineering approaches to create cardiac tissues from induced pluripotent stem cells (iPSCs). Tridimensional cell models are presented as improved models to recapitulate cardiac phenotype and functions. The topic is very interesting; however, in its current form the review does not add anything new to existing reviews from other groups.

R: We thank this Reviewer for her/his valuable inputs and suggestions.

Major comments

  1. Authors should extend all paragraphs and better cover the different facets of human cardiac tissue engineering, from the differentiation of iPSCs into CMs to CM patterning and functional maturation through the incorporation of selected biophysical and biochemical cues.

R: Thank you for your comment. As suggested, we have expanded the section to include more relevant and recent information. We discussed cardiomyocyte maturation via the incorporation of selected biophysical and biochemical cues (electrical, mechanical, co-culture, biochemical factors and ECM).

  1. How engineered cardiac constructs can help model and treat cardiac diseases should be also better discussed. Moreover, as the translational success of these constructs depends on the ability to recapitulate in vitro the complexity of functional heart, newest technological advances in molecular characterization of iPSC-CMs and prediction of their functions should be presented.

R: Thank you for your valuable suggestions. These aspects have been added to the paper.

  1. There are several studies by Orita K., Juhola M., and Joutsijoki H., among others (recently reviewed by Coronnello C., Stem Cell Reviews and Reports, 2021), demonstrating the possibility to exploit artificial intelligence techniques to assess the quality of iPSC-CMs. Authors could spend a few sentences to discuss these techniques.

R: Thank you for the suggestion. These aspects and the suggested papers were added in the revised paper.

  1. Furthermore, the review lacks of a critical evaluation. What is the authors’ view on the current status of cardiac tissue engineering? Do they have any recommendations for future research?

R: Thank you for the suggestion. Future direction was added in the conclusion section.

  1. There are several recent papers that were not included in the review. Few examples are:

Dries A M Feyen, Cell Rep, 2020 Jul 21; Willem J de Lange, Am J Physiol Heart Circ Physiol, 2021 Apr 1; Qianliang Yuan, J Cardiovasc Dev Dis, 2022 Jan; Anna Skorska, Cell Mol Life Sci, 2022 Feb.  

R: Thank you for your comment. These papers were added to the review.

Minor comments

  1. Paragraphs are often disorganized with disconnected sentences. This is particularly evident for paragraph 2. R: Thank you for your comment. Paragraphs have been re-organized in order to reconnect sentences.
  2. In paragraph 3, authors should spell out the acronym PSC. R: Thank you for your suggestion. It was corrected.
  3. ine 212. Correct "levl": R: Thank you for your suggestion. It was corrected

Reviewer 2 Report

        It is a nice short review by Varzideh et al. that summarized the recent progress in iPSC-derived engineered 3D cardiac models for the applications of disease modeling, covered the aspects of cells, biomaterials and fabrication techniques. This review is timely, in that there is an increased use of bioengineering methods to create more mature 3D cardiac models, and will be an impactful contribution to the field. It is well written, and concise. However, the authors only listed and overviewed papers and methods, and thus lacked their own thoughts on the future of this field, which from my point of view is critical for a review paper.

I have some suggestions below:

  • The methods are described one-after-another without comprehensive and deep discussions of their advantages and limitations, a table box would be very helpful.
  • Additional schematics and figures on the state-of-art techniques would helpful for the readers to understand available techniques
  • I suggest the authors to add a "outlook and future direction" section to add more insights and thoughts to the current development of such technology.

Author Response

It is a nice short review by Varzideh et al. that summarized the recent progress in iPSC-derived engineered 3D cardiac models for the applications of disease modeling, covered the aspects of cells, biomaterials and fabrication techniques. This review is timely, in that there is an increased use of bioengineering methods to create more mature 3D cardiac models, and will be an impactful contribution to the field. It is well written, and concise. However, the authors only listed and overviewed papers and methods, and thus lacked their own thoughts on the future of this field, which from my point of view is critical for a review paper.

I have some suggestions below:

  1. The methods are described one-after-another without comprehensive and deep discussions of their advantages and limitations, a table box would be very helpful.

R: Thank you for your insightful comment. We have re-organized the manuscript in order to include a discussion of advantages and limitations of the described techniques. According to this comment, we explained cardiomyocyte maturation in 2D and 3D culture.

  1. Additional schematics and figures on the state-of-art techniques would helpful for the readers to understand available techniques.

R: Thank you for your comment. An additional figure was added, as requested.

  1. I suggest the authors to add a "outlook and future direction" section to add more insights and thoughts to the current development of such technology.

R: Thank you for the valuable suggestion. A section addressing future direction was added in the conclusion section.

Round 2

Reviewer 1 Report

The authors have satisfactorily addressed my concerns. 

Reviewer 2 Report

The authors addressed all my questions.